# Single-Step Genomic Prediction of Superovulatory Response Traits in Japanese Black Donor Cows

**DOI:** 10.3390/biology12050718

**Published:** 2023-05-15

**Authors:** Atsushi Zoda, Shinichiro Ogawa, Rino Kagawa, Hayato Tsukahara, Rui Obinata, Manami Urakawa, Yoshio Oono

**Affiliations:** 1Research and Development Group, Zen-noh Embryo Transfer Center, Kamishihoro 080-1407, Japantsukahara-hayato@zennoh.or.jp (H.T.);; 2Division of Meat Animal and Poultry Research, Institute of Livestock and Grassland Science, National Agriculture and Food Research Organization (NARO), Tsukuba 305-0901, Japan

**Keywords:** Japanese Black cattle, superovulatory response trait, pedigree, single-nucleotide polymorphism, single-step genomic prediction

## Abstract

**Simple Summary:**

The embryos from Japanese Black cows are used for embryo transfer in Japan, which could contribute to obtaining a greater number of fattened progenies from dams with superior genetic abilities for the sustainable supply of high-quality Wagyu beef. We previously reported the possibility of genetically improving the in vivo embryo production performance of Japanese Black donor cows. We performed single-step genomic prediction of breeding values for superovulatory response traits in Japanese Black donor cows using genome-wide single-nucleotide polymorphism markers. The results indicate that introducing the single-step genomic prediction scheme could allow more effective selection for superovulatory response traits at a younger age of donor cows, and it might achieve more efficient genetic improvements than breeding value prediction based on only pedigree information. To the best of our knowledge, this is the first study performing single-step genomic prediction of female reproductive traits in Japanese Black cattle. In future work, an approach suitable for managing genetic diversity while continuing sound selection should be developed.

**Abstract:**

We assessed the performance of single-step genomic prediction of breeding values for superovulatory response traits in Japanese Black donor cows. A total of 25,332 records of the total number of embryos and oocytes (TNE) and the number of good embryos (NGE) per flush for 1874 Japanese Black donor cows were collected during 2008 and 2022. Genotype information on 36,426 autosomal single-nucleotide polymorphisms (SNPs) for 575 out of the 1,874 cows was used. Breeding values were predicted exploiting a two-trait repeatability animal model. Two genetic relationship matrices were used, one based on pedigree information (**A** matrix) and the other considering both pedigree and SNP marker genotype information (**H** matrix). Estimated heritabilities of TNE and NGE were 0.18 and 0.11, respectively, when using the **H** matrix, which were both slightly lower than when using the **A** matrix (0.26 for TNE and 0.16 for NGE). Estimated genetic correlations between the traits were 0.61 and 0.66 when using **H** and **A** matrices, respectively. When the variance components were the same in breeding value prediction, the mean reliability was greater when using the **H** matrix than when using the **A** matrix. This advantage seems more prominent for cows with low reliability when using the **A** matrix. The results imply that introducing single-step genomic prediction could boost the rate of genetic improvement of superovulatory response traits, but efforts should be made to maintain genetic diversity when performing selection.

## 1. Introduction

Japanese Wagyu cattle comprise four modern Japanese native cattle breeds: Japanese Black, Japanese Brown, Japanese Shorthorn, and Japanese Polled. Japanese Black cattle are the primary breed of Japanese Wagyu in Japan and are well known for their excellent meat quality, including a high degree of marbling, around the world, e.g., [1,2,3]. For representative carcass characteristics such as the degree of marbling and carcass weight, historically led by the Wagyu Registry Association in Japan, variance component estimation using the restricted (or residual) maximum likelihood (REML) approach [4,5], and then the empirical best linear unbiased prediction (BLUP) [6], of breeding values of selection candidates has been conducted in Japan with many carcass performance records of fattened progeny slaughtered at carcass markets and deep pedigree information [7,8,9]. This kind of progeny testing scheme has provided predicted breeding values for carcass traits of sire candidates with quite high reliability [9], although this scheme appears time-consuming. Recently, genomic prediction (GP) of breeding values for carcass traits has been studied in this breed, using only genotype information on commercial single-nucleotide polymorphism (SNP) markers (genomic BLUP; GBLUP) or simultaneously using SNP markers and pedigree information, namely, the single-step GBLUP (ssGBLUP) approach [10,11,12,13,14,15]. Previous studies reported that introducing breeding value prediction using genome-wide SNP markers could boost the rate of genetic gain of carcass traits, mainly by making available predicted breeding values with higher reliability at a younger age, such as pre-selection of bull and dam candidates.

Improving the reproductive efficiency of Japanese Black cattle has become increasingly important to strengthen the foundation for the domestic production of high-quality Wagyu beef in Japan [16]. The value of Japanese Black calves is much higher than that of the other beef breeds, and thus embryo transfer (ET) is now widely used in producing fattened progeny to accomplish a stable supply of high-quality Wagyu beef [17]. Recently, we estimated the genetic parameters of two in vivo embryo production-related traits using superovulatory treatments—total number of embryos and oocytes (TNE) and number of good embryos (NGE) per flush—in Japanese Black donor cows. Then, we showed that their heritabilities, estimated using a two-trait repeatability animal model with the pedigree-based additive genetic relationship matrix (**A** matrix), were higher than those at the ages of first calving and calving interval, e.g., [18,19,20], in Japanese Black cattle. This suggested the possibility of genetically improving these traits [21,22]. Furthermore, we estimated that there was a weak genetic correlation between superovulatory response traits of donor cows and carcass traits of fattened progeny, concluding that selecting donors with superior genetic ability for superovulatory responses would not be expected to have immediately antagonistic effects on carcass performance in their fattened progeny [23].

Our previous studies on genetically improving superovulatory response traits in Japanese Black cows did not use genomic information [21,22,23]. The performance of pedigree-based breeding value prediction depends on the content of pedigree data such as pedigree depth, e.g., [24,25,26]. The use of genome-wide SNP markers could capture genetic relatedness between unrelated pairs in pedigree data, e.g., [27,28,29], which might contribute to improving the reliability of breeding value prediction. When not all individuals have their own genotype information, the single-step approach would be a reasonable choice [12]. In this study, we performed GP of TNE and NGE in Japanese Black donor cows using an ssGBLUP approach. To our knowledge, this is the first study that reports the results of such an approach for female reproductive traits in Japanese Black cattle and reports those for superovulatory response traits in beef cattle.

## 2. Materials and Methods

### 2.1. Ethics Statement

Animal Care and Use Committee approval was not needed because the data were obtained from existing databases.

### 2.2. Phenotypic, SNP Genotype, and Pedigree Data

The TNE and NGE per flush were recorded from 25,332 superovulation treatments of 1874 Japanese Black donor cows, meaning the number of flushes was 13.5 per cow on average, between 2008 and 2022 at the Zen-noh Embryo Transfer Center, Hokkaido, Japan. Table 1 shows the basic statistics for phenotypic records for TNE and NGE analyzed in this study. The donor cows consisted of individuals introduced from the markets and their (grand)daughters. Most of them were also included among 1546 Japanese Black donor cows with superovulation performance records of their own used in our previous studies [21,22,23]. For detailed explanations of the definitions of TNE and NGE, see our previous study [21]. Briefly, TNE was defined as the sum of the number of embryos and unfertilized oocytes collected in a single flush, and NGE was the number of freezable embryos morphologically classified as grade 1 according to the International Embryo Transfer Society criteria. Genotype information of 575 of the 1874 Japanese Black cows was obtained using the Illumina BovineSNP50 v2 BeadChip [30]. Quality control (QC) was performed on 52,524 SNPs located on each of the 29 *Bos taurus* autosomes after updating the annotation to ARS-UCD1.2 using the LiftOver tool (https://genome.ucsc.edu/cgi-bin/hgLiftOver) (accessed on 19 October 2022) for the 52,524 SNPs. For each SNP, the criteria of QC were set as minor allele frequency > 0.01, call rate > 0.95, and *p*-value > 0.001 for the Hardy–Weinberg equilibrium (HWE) test. For each cow, the criterion of QC was set as call rate > 0.95. After quality control, missing genotypes were filled in using Beagle v5.4 [31]. Finally, genotype information on 36,426 SNPs on the Illumina BovineSNP50 v2 BeadChip was used. Pedigree data were constructed by tracing back up to five generations of the 1874 donor cows; then, the number of individuals included in the pedigree data was 4684 [30].

### 2.3. A and G Matrices

For a detailed explanation of how different relationship matrices are calculated using pedigree and/or SNP marker information via BLUPF90+ software [32], see, for instance, [12]. We obtained the block of **A** for the 575 cows with their own SNP genotypes based on pedigree data containing 4684 individuals with the saveA22 option of BLUPF90+. The genomic relationship matrix (**G** matrix) based on genotype information on 36,426 SNPs, which were calculated using VanRaden’s method l [33], was obtained using the saveG option of BLUPF90+. Allele frequencies of the 36,426 SNPs were calculated using the 575 cows.

The pedigree-based inbreeding coefficients for the 575 cows were calculated as the diagonal elements of the block of **A** minus 1. The genome-based inbreeding coefficients were calculated as the diagonal elements of **G** minus 1. The kinship coefficient between cows *i* and *j* (*R_ij_*) was calculated as follows:$$R_{ij}=\frac{a_{ij}}{\sqrt{a_{ii}}\sqrt{a_{jj}}},$$
where *a_ij_* is the (*i*,*j*) element of **A** for calculating pedigree-based kinship coefficient and **G** for calculating genomic kinship coefficient. Note that the **A** matrix is twice the matrix of identity-by-descent probabilities, while the **G** matrix is twice the matrix of identity-by-state probabilities under the allele frequencies used to obtain the **G** matrix [34].

### 2.4. Variance Component Estimation

The following two-trait repeatability animal model was used [21]:$$\left[\begin{array}{c} y_{TNE}\\ y_{NGE} \end{array}\right]=\left[\begin{array}{cc} X_{TNE} & 0\\ 0 & X_{NGE} \end{array}\right]\left[\begin{array}{c} b_{TNE}\\ b_{NGE} \end{array}\right]+\left[\begin{array}{cc} Z_{TNE} & 0\\ 0 & Z_{NGE} \end{array}\right]\left[\begin{array}{c} a_{TNE}\\ a_{NGE} \end{array}\right]+\left[\begin{array}{cc} W_{TNE} & 0\\ 0 & W_{NGE} \end{array}\right]\left[\begin{array}{c} pe_{TNE}\\ pe_{NGE} \end{array}\right]+\left[\begin{array}{c} e_{TNE}\\ e_{NGE} \end{array}\right]$$
where y_i_ is the vector of phenotypic records for trait i; b_i_, a_i_, pe_i_, and e_i_ are the vectors of fixed effects (year of superovulation, month of superovulation, type of superovulation program, technician, and linear and quadratic covariates of age in months at superovulation), breeding values, permanent environmental effects, and errors; and X_i_, Z_i_, and W_i_ are the design matrices relating y_i_ to b_i_, a_i_, and pe_i_, respectively. When performing pedigree-based breeding value prediction, the (co)variance structure was as follows:$$\mathrm{var}\left[\begin{array}{c} a_{TNE}\\ a_{NGE}\\ pe_{TNE}\\ pe_{NGE}\\ e_{TNE}\\ e_{NGE} \end{array}\right]=\left[\begin{array}{cccccc} A\sigma_{aTNE}^{2} & A\sigma_{aTNE,NGE} & 0 & 0 & 0 & 0\\ A\sigma_{aTNE,NGE} & A\sigma_{aNGE}^{2} & 0 & 0 & 0 & 0\\ 0 & 0 & I\sigma_{peTNE}^{2} & I\sigma_{peTNE,NGE} & 0 & 0\\ 0 & 0 & I\sigma_{peTNE,NGE} & I\sigma_{peNGE}^{2} & 0 & 0\\ 0 & 0 & 0 & 0 & I\sigma_{eTNE}^{2} & I\sigma_{eTNE,NGE}\\ 0 & 0 & 0 & 0 & I\sigma_{eTNE,NGE} & I\sigma_{eNGE}^{2} \end{array}\right]$$
where *σ*^2^*_ai_* is the additive genetic variance of trait *i*; *σ*_*a*12_ is the additive genetic covariance between traits; *σ*^2^*_pei_* is the permanent environmental variance of trait *i*; *σ*_*pe*12_ is the permanent environmental covariance between traits; *σ*^2^*_ei_* is the error variance for trait *i*; *σ*_*e*12_ is the error covariance between traits; and **A** is the additive genetic relationship matrix for 4684 animals in the whole pedigree data. When performing the analyses using genomic information, the **A** matrix was replaced by a matrix **H** calculated as follows [10,11,12]:$$\begin{array}{cc} \; & H=\left[\begin{array}{cc} A_{11}-A_{12}A_{22}^{-1}A_{21}+A_{12}A_{22}^{-1}G^{*}A_{22}^{-1}A_{21} & A_{12}A_{22}^{-1}G^{*}\\ G^{*}A_{22}^{-1}A_{21} & G^{*} \end{array}\right]\\ \; & \;\;\;\;\;=\left[\begin{array}{cc} A_{11} & A_{12}\\ A_{21} & A_{22} \end{array}\right]+\left[\begin{array}{cc} A_{12}A_{22}^{-1}\left(G^{*}-A_{22}\right)A_{22}^{-1}A_{21} & A_{12}A_{22}^{-1}\left(G^{*}-A_{22}\right)\\ \left(G^{*}-A_{22}\right)A_{22}^{-1}A_{21} & \left(G^{*}-A_{22}\right) \end{array}\right]\\ \; & \;\;\;\;\;=A+\left[\begin{array}{cc} A_{12}A_{22}^{-1}\left(G^{*}-A_{22}\right)A_{22}^{-1}A_{21} & A_{12}A_{22}^{-1}\left(G^{*}-A_{22}\right)\\ \left(G^{*}-A_{22}\right)A_{22}^{-1}A_{21} & \left(G^{*}-A_{22}\right) \end{array}\right] \end{array}\;$$
where **G*** = 0.95 **G** + 0.05 **A**_22_, and **A**_22_ is the block of **A** for the 575 cows with their own genotypes. Then, the inverse of the **H** matrix (**H**^−1^) can be obtained as follows [35,36]:$$\begin{array}{cc} \; & H^{-1}=\left[\begin{array}{cc} {\left(A_{11}-A_{12}A_{22}^{-1}A_{21}\right)}^{-1} & -{\left(A_{11}-A_{12}A_{22}^{-1}A_{21}\right)}^{-1}A_{12}A_{22}^{-1}\\ -A_{22}^{-1}A_{21}{\left(A_{11}-A_{12}A_{22}^{-1}A_{21}\right)}^{-1} & A_{22}^{-1}+A_{22}^{-1}A_{21}{\left(A_{11}-A_{12}A_{22}^{-1}A_{21}\right)}^{-1}A_{12}A_{22}^{-1} \end{array}\right]+\left[\begin{array}{cc} 0 & 0\\ 0 & G^{*-1}-A_{22}^{-1} \end{array}\right]\\ \; & \;\;\;\;\;\;\;\;\;=A^{-1}+\left[\begin{array}{cc} 0 & 0\\ 0 & G^{*-1}-A_{22}^{-1} \end{array}\right] \end{array}\;$$

Variance components were estimated via the average information-restricted maximum likelihood (AI-REML) algorithm with the method VCE option of BLUPF90+.

### 2.5. Breeding Value Prediction

Using the BLUPF90+ software, breeding values were predicted by changing the relationship matrix (two patterns: **A** or **H**) and the variance components (two patterns: those estimated using either **A** or **H**). Therefore, four patterns were considered to predict breeding values in this study. The reliability of predicted breeding values for cow *i* (*rel_i_*) was calculated as follows:$$rel_{i}=1-\frac{PEV_{i}}{a_{ii}\sigma_{a}^{2}}$$
where *a_ii_* is the *i*th diagonal element of the relationship matrix used for breeding value prediction (**A** or **H**); *σ*^2^*_a_* is the additive genetic variance estimated using either **A** or **H**; and *PEV_i_* is the prediction error variance of breeding value for cow *i* under one of the four settings.

### 2.6. Selection Simulation

To collect information on implementing a genomic selection scheme on the studied population, 25 cows with top predicted breeding value for TNE were selected from the 575 cows, assuming that the 575 cows were selection candidates. First, the degree of consistency of the selected 25 cows was evaluated among the four patterns of breeding value prediction. Next, the mean kinship coefficient, both pedigree-based and genome-based, among the selected 25 cows was calculated. For comparison, an empirical distribution of the mean kinship coefficient was prepared by calculating the mean kinship coefficient among 25 randomly selected cows. The number of iterations to obtain the empirical distribution was 1,000,000. Note that pedigree-based kinship coefficients used here were calculated using pedigree information on all available 4684 animals.

## 3. Results and Discussion

### 3.1. Comparing A and G Matrices for the 575 Cows

Figure 1 shows a histogram of MAFs in the 575 cows for the 36,426 SNP markers used in the ssGBLUP approach. The shape of the histogram is similar to those in previous studies on different Japanese Black cattle populations [37,38,39]; that is, the proportion of SNPs with low MAFs was slightly greater than that of the others. The ET center introduces cows from various parts of Japan to meet a wide demand of embryo buyers, which might confer results on the MAF distribution similar to those in previous studies.

Table 2 summarizes the basic statistics for inbreeding and kinship coefficients. The pedigree-based inbreeding coefficients of the 575 cows were more discrete than the genomic inbreeding coefficients. Values of genomic inbreeding coefficients were <0 for some cows (Figure 2). Pearson’s correlation coefficient between the two inbreeding coefficients was 0.03. Conceptually, the pedigree-based inbreeding coefficient increases with greater homozygosity regardless of allele type (major or minor), while the genomic inbreeding coefficient increases or decreases with allele type because the values of allele frequencies used for calculating the **G** matrix are not fixed at 0.5. The cows with pedigree-based inbreeding coefficients of 0 have various values of genomic inbreeding coefficients. Some of the sires of the donor cows with pedigree-based inbreeding coefficients >0 were also the maternal grand sires of the cows.

Pearson’s correlation coefficient between pedigree-based and genomic kinship coefficients was 0.79, while genomic kinship coefficients took more continuous values (Figure 3). This would be because **G** can capture the information on Mendelian sampling. A greater value of pedigree-based kinship coefficient reflects smaller variation in the value of the genomic kinship coefficient. This could be because a more distant pedigree-based kinship is associated with a greater number of generations or number of occurrences of Mendelian sampling events between the cows. In this study, pairs with pedigree-based kinship coefficients >0 included mother–daughter pairs, full-sib pairs, and half-sib pairs. Komiya et al. [38] compared pedigree-based kinship coefficients and the number of opposing homozygotes in a Japanese Black cattle population and found that, as pedigree-based kinship coefficients decreased, the number of opposing homozygotes and its variation tended to increase. The use of genomic information gave a non-zero value as a kinship coefficient for pairs with a pedigree-based kinship coefficient of 0. This would make an individual with a pedigree-based kinship coefficient of 0 a source of additional information in GP of breeding values.

### 3.2. Variance Component Estimation

Estimated heritability when using the **A** matrix was 0.26 for TNE and 0.16 for NGE, and their genetic correlation was estimated to be 0.66. With somewhat small datasets, previous studies obtained similar results [21,23]. Estimated heritability was slightly lower when using the **H** matrix, while there was almost no difference in estimated repeatability and genetic correlation when using **A** and **H** matrices (Table 3). This suggests that the problem of missing heritability, if any, might be small. Ogawa et al. [40] compared genetic parameters for carcass weight and marbling score in Japanese Black cattle estimated with **G** matrices calculated using 31,231 and 565,837 SNP markers and found that the difference in values of estimated heritability was small, although the effect of chip ascertainment bias was unknown. Takeda et al. [41] reported that the estimated heritabilities for carcass weight and feed efficiency traits were almost the same or slightly lower when changing from **A** matrix to **H** matrix in variance component estimation. In both Japanese Black cattle and Holstein cattle, Nagai et al. [42] estimated lower heritability for semen production traits using the **G** matrix than those estimated using the **A** matrix, while the estimated repeatabilities were similar to each other. Reasons for lower heritability using genomic information in this study may be the small number of genotyped animals and that the studied traits are highly polygenic. On the other hand, another possible reason may be that the allele frequencies used to construct the **G** matrix were calculated using the 575 cows, which would be different from those in the base population. Indeed, representative carcass characteristics of Japanese Black fattened progenies have been genetically improved by selecting elite sires [9]; however, we previously estimated the genetic correlations between carcass traits and superovulatory response traits to be weak to negligible. On the other hand, Onogi et al. [14] reported that heritability estimates for marbling score, carcass weight, and ribeye area varied depending on the particular parameter settings. Therefore, in the future, the effect of different parameter settings on genetic parameter estimation and breeding value prediction for superovulatory response traits in Japanese Black cattle should be assessed.

### 3.3. Reliability of Predicted Breeding Values

The mean and standard deviation of reliabilities of predicted breeding values are summarized in Table 4. Figure 4 compares the reliabilities among different relationship matrices (**A** or **H**) used in predicting breeding values and different variance components (estimated using **A** or **H**). As already reported in [21], reliability increased with the number of records. For both traits, when the same relationship matrix was used in predicting breeding values, the reliabilities were, on average, greater when variance components estimated with the **A** matrix were used than when using those estimated with the **H** matrix. This would be because the estimated heritability was higher when using the **A** matrix than when using the **H** matrix (Table 3). When the variance components were the same, the reliabilities were, on average, greater when using the **H** matrix in predicting breeding values than when using the **A** matrix. This might be because the **G** matrix, which was used to construct the **H** matrix, had more non-zero non-diagonal elements than the **A** matrix. Figure 5 shows the differences between the reliabilities when using **A** and **H** matrices in predicting breeding values. The difference in values of reliabilities was more prominent when the number of records was smaller. This result indicates that the ssGBLUP approach would be more effective in selecting candidates at a younger age, such as at pre-selection, than traditional pedigree-based breeding value prediction. It would be important to increase the number of genotyped individuals to further improve the reliability of genomic breeding value for the superovulatory response trait. Note that the setting of the **H** matrix for REML and **A** matrix for BLUP never happens in practice.

### 3.4. Selection Simulation

Table 5 lists the Pearson and Spearman rank correlations between breeding values predicted under the different settings for each trait. For both traits, values of correlation coefficients were highly positive (>0.9). This indicates that selection based on different predicted breeding values would lead to the choice of similar cows but not the same exact cows (Table 6).

Figure 6 shows the empirical distributions of the mean pedigree-based and genomic kinship coefficients among 25 cows randomly selected from the 575 cows. The distributions were positively skewed for both pedigree-based and genomic kinship coefficients. Table 7 summarizes the mean kinship coefficients among 25 cows selected based on predicted breeding values for TNE and the *p*-values of the mean kinship coefficients calculated using the empirical distributions. Regardless of the source of information (pedigree-based or genomic), the smallest *p*-value was obtained when the matrices for breeding value prediction and variance component estimation were the **H** and **A** matrices, respectively. Meanwhile, the *p*-value was largest when the matrices for breeding value prediction and variance component estimation were the **A** and **H** matrices, respectively. Interestingly, the *p*-value for the pedigree-based kinship coefficient was smaller when the matrices for breeding value prediction and variance component estimation were both the **H** matrix, while that for the genomic kinship coefficient was smaller when the matrices for breeding value prediction and variance component estimation were both the **A** matrix. These findings might have been due to the pedigree-based and genomic kinship coefficients being able to reflect expected and realized relationships, respectively.

### 3.5. General Discussion

For Japanese Black cattle, studies on GP of carcass traits, e.g., [13,14,15], and other traits, including feed efficiency, meat quality, and semen production, have been performed e.g., [41,42,43,44]. Here, we assessed the performance of the ssGBLUP approach in predicting breeding value for superovulatory response traits in Japanese Black donor cows. Our results show that more reliable prediction could be achieved using ssGBLUP than the conventional BLUP approach based on only pedigree information, especially when the number of records is small (Figure 5). Generally, improving reproductive efficiency is difficult due to the late availability of records. Zoda et al. [21] stated that, by applying the BLUP approach, the prediction reliability was around 0.6 when the number of repeated records was 10, but it increased only slowly beyond this. Figure 4 shows that, under the same variance components, the value of average reliability using the ssGBLUP approach when the number of repeated records was five or more was equal to or greater than that using BLUP when the number of records was 10. This indicates that introducing the ssGBLUP approach could contribute to increasing the rate of genetic gain by shortening the generation interval. To the best of our knowledge, this is the first study reporting GP of female reproductive traits in Japanese Black cattle using the ssGBLUP approach. For GP of superovulatory response traits, only Cornelissen et al. [45] reported the results of the ssGBLUP approach in Holstein dairy cattle, and there are no reports for beef cattle. Therefore, this study should provide important information on efficient genetic improvement of in vivo embryo production in cattle.

The advantages of the ssGBLUP approach over BLUP include that the relationship matrix for covariance structure of breeding values is denser and that information on Mendelian sampling is accounted for. Komiya et al. [38] illustrated a negative relationship between the kinship coefficient based on deep pedigree information and the number of opposing homozygotes on SNP markers in Japanese Black cattle genotyped using the Illumina 50K chip. Zoda et al. [39,46] reported that the results of **G** matrix calculation and STRUCTURE analysis using SNP markers genotyped using the Illumina 50K chip reflected the population structure of Japanese Black cattle. These findings indicate the possibility of assessing the genetic similarity among Japanese Black cattle using commercial SNP markers, although there are concerns about marker density and SNP ascertainment bias. Recently, Fujii et al. [47] performed a study on preimplantation genomic selection in Japanese Black cattle. Preimplantation genomic selection might have the potential to further increase the rate of genetic gain when combined with superovulation treatment.

In dairy cattle populations, the introduction of a genomic selection scheme accelerated the rate of decline in genetic diversity, e.g., [48,49,50]. Considering the results shown in Table 7, further study needs to be conducted to develop a methodology to properly manage the genetic diversity of this population while continuing genetic selection, e.g., [51,52,53]. Using pedigree analysis, Nomura et al. [54] found a sharp decline in the effective population size of Japanese Black cattle mainly due to the intensive use of a small number of elite sires with high predicted breeding value for meat quality. Therefore, the development of a suitable method is valuable for evaluating and controlling the genetic diversity of the Japanese Black cattle population using genomic information, e.g., [39,55,56].

The pedigree information used in this study could be shallow because of limited information on the ancestry of the externally introduced cows [30]. The performance of breeding value prediction using pedigree information depends on the completeness (richness) of the pedigree, e.g., [26,57,58]. Incomplete or missing pedigrees could affect the performance of single-step prediction [59]. To address this, the use of a metafounder, e.g., [60,61,62], or a GP approach that does not use pedigree information in the future needs to be considered.

In Holstein cattle populations, genome-wide association studies (GWASs) have been conducted for in vivo embryo production traits and plasma anti-Müllerian hormone levels using commercial SNP markers [63,64,65]. In Japanese Black cattle, GWASs have been performed for several female reproductive traits, including age at first calving, days open, and the number of inseminations to conception [66,67,68,69,70]. Hirayama et al. [71] examined the association of superovulation response with polymorphisms of follicle-stimulating hormone receptor (FSHR) and glutamate ionotropic receptor AMPA-type subunit 1 (GRIA1) genes, but no GWASs have been performed for superovulatory response traits. Our population could be used to conduct GWASs for superovulatory response traits in Japanese Black cattle. To increase the power of GWASs, it is worth considering re-genotyping [30] for imputation to higher-density marker data as well as increasing the number of cows with both phenotype and genotype data in the future.

## 4. Conclusions

The embryos from Japanese Black cows are used for embryo transfer, which could contribute to producing fattened progenies for the stable supply of high-quality Wagyu beef. Here, we performed GP of superovulatory response traits, namely, TNE and NGE, in Japanese Black donor cows using the ssGBLUP approach and approximately 36,000 commercially available autosomal SNP markers. The main finding was that, when the variance components were the same in breeding value prediction, the mean reliability was greater when using the **H** matrix than when using the **A** matrix. This was more prominent for the cows with low reliability when using the **A** matrix. Therefore, introducing the ssGBLUP approach could boost the rate of genetic gain of superovulatory response traits in this population. Meanwhile, there is a need to develop an approach to manage genetic diversity while continuing sound selection.

## Figures and Tables

**Figure 1 biology-12-00718-f001:**
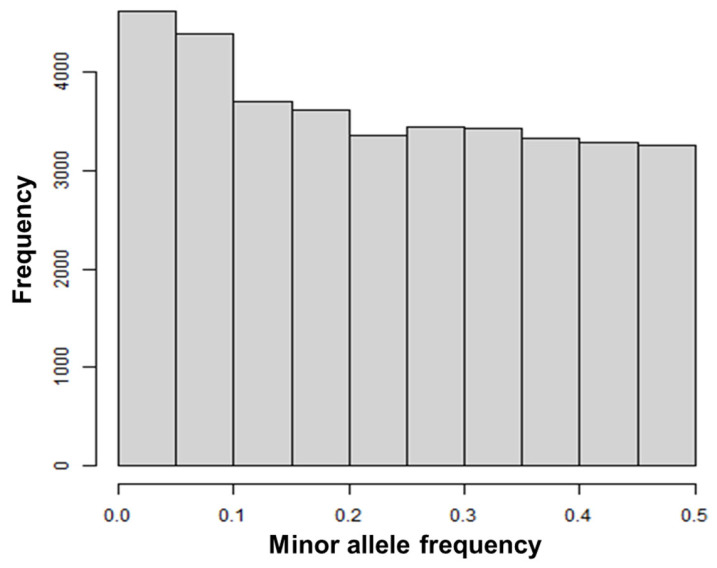
Histogram of minor allele frequencies for 575 Japanese Black donor cows with 36,426 single-nucleotide polymorphism markers.

**Figure 2 biology-12-00718-f002:**
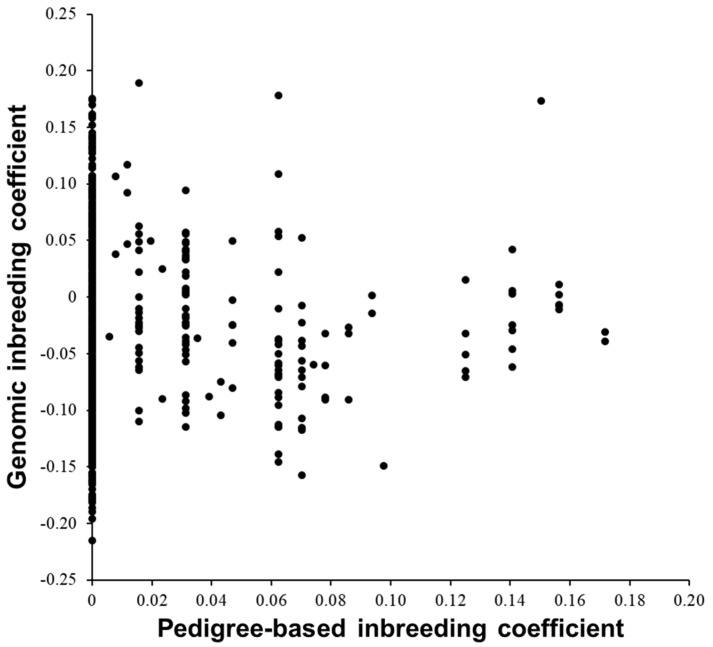
Scatterplot of pedigree-based and genomic inbreeding coefficients of 575 Japanese Black donor cows.

**Figure 3 biology-12-00718-f003:**
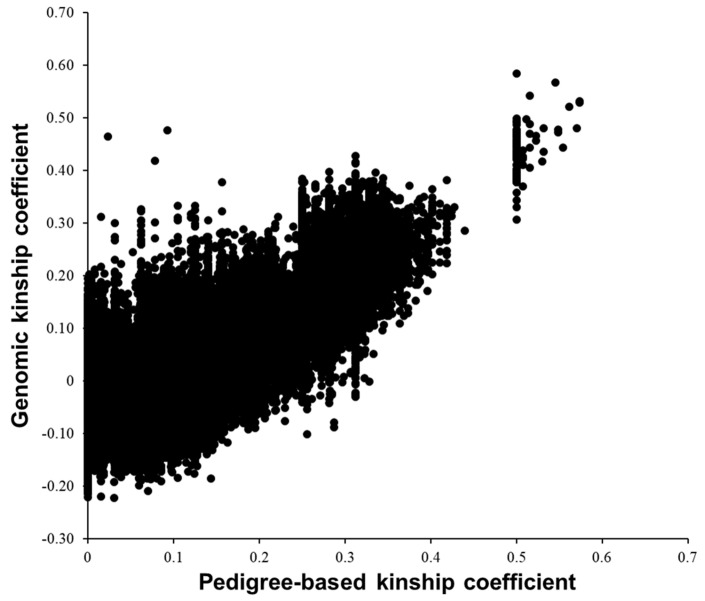
Scatterplot of pedigree-based and genomic kinship coefficients among 575 Japanese Black donor cows.

**Figure 4 biology-12-00718-f004:**
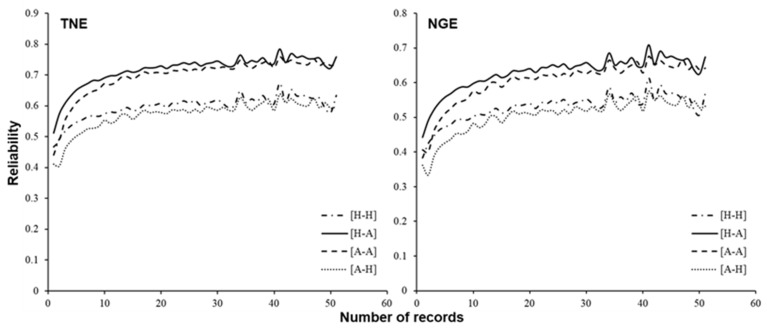
Relationships between the number of superovulatory response records per cow and the average reliability of predicted breeding values for 575 cows. H–H: Matrices for both BLUP and REML were the **H** matrix; H–A: Matrix for BLUP was the **H** matrix and that for REML was the **A** matrix; A–H: Matrix for BLUP was the **A** matrix and that for REML was the **H** matrix; A–A: Matrices for BLUP and REML were both the **A** matrix.

**Figure 5 biology-12-00718-f005:**
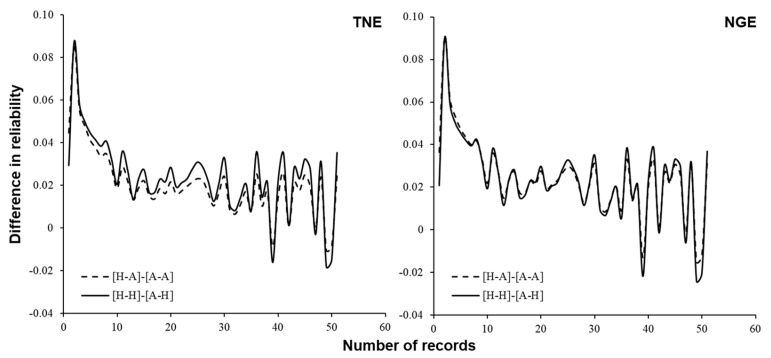
Relationships between the number of superovulatory response records per cow and the difference of average reliability of predicted breeding values for 575 cows between the results obtained using different matrices for BLUP. H–H: Matrices for BLUP and REML were both the **H** matrix; H–A: Matrix for BLUP was the **H** matrix and that for REML was the **A** matrix; A–H: Matrix for BLUP was the **A** matrix and that for REML was the **H** matrix; A–A: Matrices for BLUP and REML were both the **A** matrix.

**Figure 6 biology-12-00718-f006:**
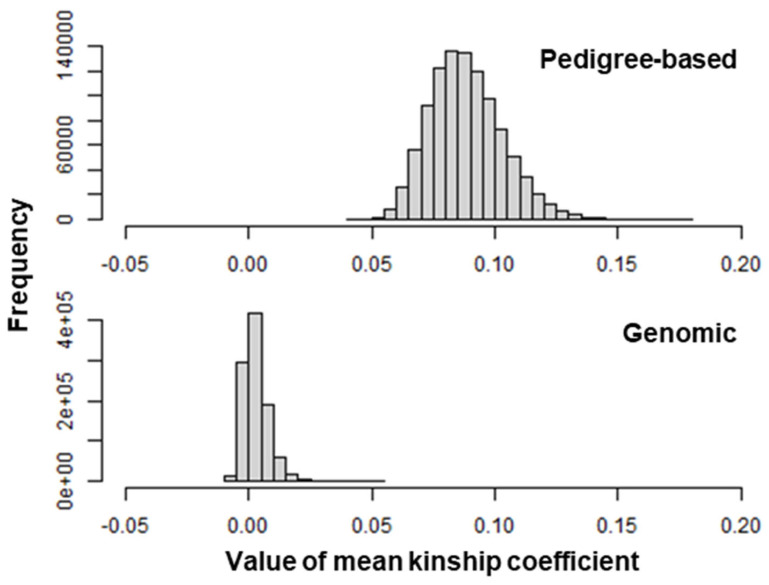
Empirical distributions of mean pedigree-based and genome-based kinship coefficient among 25 cows randomly selected from 575 Japanese Black donor cows.

**Table 1 biology-12-00718-t001:** Basic statistics for 25,332 phenotypic records obtained from 1874 donor cows.

Trait Name	Abbreviation	Mean	Standard Deviation
Total number of embryos and oocytes	TNE	16.3	10.4
Number of good embryos	NGE	6.7	6.0

**Table 2 biology-12-00718-t002:** Basic statistics for inbreeding and kinship coefficients for 575 Japanese Black donor cows.

Coefficient	Mean	Standard Deviation	Minimum	Maximum
Pedigree-based inbreeding	0.014	0.032	0	0.172
Genomic inbreeding	–0.033	0.080	–0.215	0.189
Pedigree-based kinship	0.087	0.089	0	0.574
Genomic kinship	0.003	0.090	–0.222	0.584

**Table 3 biology-12-00718-t003:** Estimated genetic parameters ± their standard errors.

Item	H Matrix		A Matrix	
	TNE	NGE	TNE	NGE
Additive genetic variance	19.19 ± 3.20	3.99 ± 0.79	28.07 ± 4.15	5.57 ± 1.05
Permanent environmental variance	17.49 ± 2.23	4.93 ± 0.58	10.87 ± 2.78	3.63 ± 0.73
Error variance	68.16 ± 0.63	26.32 ± 0.24	68.15 ± 0.63	26.32 ± 0.24
Heritability	0.18 ± 0.03	0.11 ± 0.02	0.26 ± 0.03	0.16 ± 0.03
Repeatability	0.35 ± 0.01	0.25 ± 0.01	0.36 ± 0.01	0.26 ± 0.01
Genetic correlation	0.61 ± 0.08		0.66 ± 0.07	

**Table 4 biology-12-00718-t004:** Basic statistics for reliabilities of predicted breeding values for 575 Japanese Black donor cows.

Matrix for BLUP	Matrix for REML	TNE		NGE	
		Mean	Standard Deviation	Mean	Standard Deviation
**H** matrix	**H** matrix	0.592	0.002	0.523	0.002
**H** matrix	**A** matrix	0.713	0.002	0.623	0.002
**A** matrix	**H** matrix	0.566	0.002	0.498	0.003
**A** matrix	**A** matrix	0.692	0.002	0.597	0.003

**Table 5 biology-12-00718-t005:** Pearson and Spearman rank correlation coefficients (upper and lower diagonals) of predicted breeding values among different settings.

Matrix for BLUP	Matrix for REML	Matrix for BLUP			
		H Matrix		A Matrix	
		Matrix for REML			
		H Matrix	A Matrix	H Matrix	A Matrix
		*TNE*			
**H** matrix	**H** matrix	--	0.990	0.972	0.959
**H** matrix	**A** matrix	0.990	--	0.974	0.979
**A** matrix	**H** matrix	0.970	0.971	--	0.990
**A** matrix	**A** matrix	0.956	0.976	0.989	--
		*NGE*			
**H** matrix	**H** matrix	--	0.990	0.966	0.954
**H** matrix	**A** matrix	0.989	--	0.964	0.971
**A** matrix	**H** matrix	0.961	0.958	--	0.989
**A** matrix	**A** matrix	0.950	0.967	0.988	--

**Table 6 biology-12-00718-t006:** Concordance rate of 25 cows selected from 575 Japanese Black donor cows for TNE and NGE (upper and lower diagonals) according to different criteria.

**Matrix for BLUP**	**Matrix for REML**	**Matrix for BLUP**			
		**H Matrix**		**A Matrix**	
		**Matrix for REML**			
		**H Matrix**	**A Matrix**	**H Matrix**	**A Matrix**
**H** matrix	**H** matrix	--	88%	92%	88%
**H** matrix	**A** matrix	88%	--	84%	84%
**A** matrix	**H** matrix	80%	68%	--	96%
**A** matrix	**A** matrix	76%	72%	80%	--

**Table 7 biology-12-00718-t007:** Mean kinship coefficient among 25 cows selected.

Matrix for BLUP	Matrix for REML	Mean Kinship Coefficient		*p*-Value Based on Empirical Distribution	
		Pedigree-Based	Genomic	Pedigree-Based	Genomic
**H** matrix	**H** matrix	0.109	0.006	0.009	0.225
**H** matrix	**A** matrix	0.110	0.009	0.008	0.113
**A** matrix	**H** matrix	0.105	0.006	0.135	0.237
**A** matrix	**A** matrix	0.105	0.006	0.130	0.199

## Data Availability

The data supporting the findings of this study are shown in the manuscript.

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
