# Peer review of "Single-Step Genomic Prediction of Superovulatory Response Traits in Japanese Black Donor Cows"

_biology, 2023, doi:10.3390/biology12050718_

Round 1

Reviewer 1 Report

This is an interesting and well-written manuscript aimed to perform genomic prediction for superovulatory response traits in Japanese Black donor cows. The methodology for genomic prediction is described with sufficient detail, but more information about GWAS appears to be needed. Results are clear and Discussion is well supported. Conclusions are interesting.

I suggest considering next minor comments to improve the manuscript:

-       What was de genomic model used for GWAS analyses?

-       What were the assumptions of the genomic model?

-       How many SNPs resulted as significant for TNE and NGE traits?

-       Did the SNPs complied with Bonferroni adjustment to test for multiple comparisons?

-       I suggest to include more information in Results and Discussion section about GWAS results for TNE and NGE traits.

-       References within the text should be in square brackets, I suggest to remove round brackets.

Author Response

Dear reviewer1.

First of all, I would like to thank you very much for your professional review. I have carefully read every question you have asked and carefully revised it one by one. Every question you have asked is very valuable. Please see the attachment and kindly review it again.

Best wishes,

Atsushi Zoda

Reviewer 2 Report

General comments

The manuscript is written well. And the subject is interesting and useful in practice.

The study focuses more on A and H matrices than on inverses of A and H matrices. However, statistically speaking, those inverses are more important and informative when comparing estimates and reliabilities.

The definitions of A and G are different, so a simple comparison of the distributions is useless (as in Figure 6 and Table 7). Once both matrices are inverted, they are informative and comparable.

According to the results in this study, predicted breeding values are more accurate and genetic selection is more efficient when using A than H (or G) for those traits. This is likely due to a small number of genotyped animals, which cannot capture sufficient genomic information.

Introduction

It should be briefly defined here and/or in Materials and Methods what “good embryos” is.

Materials and Methods

I am not sure if parity is more important than age in the model. The same age effects in different parities may be significantly different. If so, age nested in parity in the model would be better.

In variance component estimation (VCE), we do not use the H matrix but the inverse of H. It is much more important to show the H inverse instead of the H matrix (just like we do not use A but the A inverse in VCE and BLUP). Therefore, authors should cite papers showing H [10] (Legarra et al., 2009) and H-inverse (Aguilar et al., 2010) instead of [7-9] and this formula of H=…..

In BLUP, again, we do not use A or H but the inverse of A or H.

Results

What was the correlation between inbreeding coefficients in A and G?

The correlation between A and G was positive. How much?

Discussion

In genetic diversity study, all available animals excluding genotyped animals with no phenotypes or progeny should be used with or without genotypes, especially when genotyping only highly selected animals, which do not reflect the current whole population.

Collecting more genotyped individuals is much more important than collecting higher density marker data in genomic selection for polygenic traits.

Conclusions

“when the variance components were the same in breeding value prediction, the mean reliability was greater when using the H matrix than when using the A matrix” => This result in Table 4 is likely due to less genomic information in G (or H), which means estimated variance components are biased and not reliable. Reliability in predicted breeding values depends on heritability and inverses of A and G when using the same pedigrees and phenotypes. In Table 4, without empirical BLUP, variance components used in BLUP should be the same as in REML. In empirical BLUP, estimated variance components in REML can be used in BLUP as known parameters (i.e., H in BLUP and A in REML). A in BLUP and H in REML never happens in practice.

References

[6] and [7] are not the first papers describing the Average Information REML. Gilmour et al. (1995) in Biometrics and Johnson and Thompson (1995) in JDS should be the ones.

Table 1:

It is better to include numbers of records (25332) and cows (1874) in the table.

Does it mean 13.5 records(flushes) per cow on average?

Table 3:

The smaller additive genetic variance and lower heritability as well as lower reliability with H are likely due to smaller diagonals of H inverse than those of A inverse, which means less genomic information than pedigree information. It is hard to see this in A and H, but if we look at inverses of A and H, we will see the difference. One reason of less genomic information could be the small number of genotyped animals: in this case, 575 cows. Another reason could be that the traits are highly polygenic, meaning controlled by many genes (probably tens or hundreds or even more) that have very small genetic effects.

It is easy to read as a scientific paper.

Author Response

Dear reviewer 2.

First of all, I would like to thank you very much for your professional review. I have carefully read every question you have asked and carefully revised it one by one. Every question you have asked is very valuable. Please see the attachment and kindly review it again.

Best wishes,

Atsushi Zoda

Round 2

Reviewer 2 Report

"... a matrix H calculated as follows [10-12]" should be "... [11]", so remove [12] and [10]. Misztal et al. (2009) from references because the formula H = ... is wrong. The correct formula is in Legarra et al. (2009).

Also, the formula H = ... is correct in Legarra et al. (2009), but the paper doesn't show H inverse = .... So, [11] should be removed from "... the inverse of H matrix (H-1) can be obtained as follows [11,35]". About the H inverse formula, two papers were published at the same time: Christensen and Lin (2009) and Aguilar et al. (2009).

Please check all other references carefully.

Author Response

We would like to appreciate the reviewer’s two rounds of review deeply. The comments and suggestions improved our manuscript considerably.

"... a matrix H calculated as follows [10-12]" should be "... [11]", so remove [12] and [10]. Misztal et al. (2009) from references because the formula H = ... is wrong. The correct formula is in Legarra et al. (2009).

Also, the formula H = ... is correct in Legarra et al. (2009), but the paper doesn't show H inverse = .... So, [11] should be removed from "... the inverse of H matrix (H-1) can be obtained as follows [11,35]". About the H inverse formula, two papers were published at the same time: Christensen and Lin (2009) and Aguilar et al. (2009).

Please check all other references carefully.

Response:

Thank you for your meaningful comment. We have revised the reference (Line 156) and checked other references.